# Sentence Prosody and Register Variation in Arabic

## Sam Hellmuth 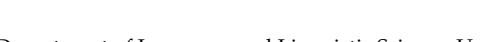

Department of Language and Linguistic Science, University of York, Heslington, York YO10 5DD, UK;
sam.hellmuth@york.ac.uk

**Abstract:** Diglossia in Arabic differs from bilingualism in functional differentiation and mode of acquisition of the two registers used by all speakers raised in an Arabic-speaking environment. The 'low' (L) regional spoken dialect is acquired naturally and used in daily life, but the 'high' (H) variety, Modern Standard Arabic, is learned and used in formal settings. Register variation between the two ends of this H–L continuum is ubiquitous in everyday interaction, such that authors have proposed distinct intermediate register levels, despite evidence of mixing of H and L features, within and between utterances, at all linguistic levels. The role of sentence prosody in register variation in Arabic is uninvestigated to date. The present study examines three variables (F0 variation, intonational choices and post-lexical utterance-final laryngealization) in 400+ turns at talk produced by one speaker of San'ani Arabic in a 20 min sociolinguistic interview, coded for register on three levels: formal (*fusħa*), 'middle' (*wusṭaː*) and dialect (*ʕaːmijja*). The results reveal a picture of key shared features across all register levels, alongside distinct properties which serve to differentiate the registers at each end of the continuum, at least some of which appear to be under the speaker's control.

**Keywords:** Modern Standard Arabic; San'ani Arabic; diglossia; multilingualism; prosody; F0

## 1. Introduction

### 1.1. Diglossia in Arabic

The Arabic language situation is a classic, and perhaps unique, example of diglossia, with speakers alternating between a 'low' spoken regional variety (L), acquired naturally, and a 'high' variety (H), Modern Standard Arabic (MSA), learned and used in formal settings (Ferguson 1959). Mastery of MSA as well as dialect is part of what it means to be a "socially competent" speaker of Arabic (Khamis-Dakwar and Froud 2019, p. 300). Acceptance of this stance is reflected in the increasing switch towards integrated approaches to teaching Arabic as a foreign language so that learners how to use both dialect and MSA, and in the process also learn when to use them (Younes 2014).

The classic characterization of diglossia in Ferguson (1959) distinguishes diglossia from both bilingualism and from a 'standard-with-dialects' model. In bilingualism, the learner acquires two languages which are structurally distinct but which can both be used in the same situations. In a standard-with-dialects context, a learner acquires two varieties of the same language which are used in different situations, but for some speakers the standard variety is their dialect. In diglossia, the learner acquires two varieties which are used in different situations, and the two varieties share enough linguistic features to be recognized as the 'same' language, despite differing in many ways; crucially, however, the standard variety is not the dialect of any speakers. Ferguson defined the H and L varieties in diglossia in terms of fundamental differences in their functional distribution, prestige, literary heritage, mode of acquisition and degree of standardization.

The practical reality is complex, however, with most speakers operating comfortably on a range of levels (Bassiouney 2009). A number of authors have therefore conceptualized the H–L distinction in terms of multiple levels, ranging from three (Mitchell 1984, 1986) to nine (Parkinson 1991), with five levels commonly proposed (e.g., Badawi 1973). For example, Mitchell (1984) identified three distinct levels by supplementing the basic

H (formal)–L (informal) divide, with a further subdivision of the informal register into 'careful' versus 'casual'. This middle level, often referred to in Arabic as *wusṭa:* 'middle', is sometimes characterized as the form used in conversations between Arabs from different dialect backgrounds. In this communicative context, local variants which are unlikely to be accessible to those outside the relevant speech community are avoided, and replaced with words or linguistic features which are shared across spoken dialects, and may indeed also be found in MSA.

Despite the practical utility of conceiving of register variation in terms of levels, it is increasingly accepted that the formal and colloquial varieties do not form a dichotomy, but lie instead at opposite ends of a continuum of variation between MSA and spoken dialects (Mejdell 2019). Recent neurophysiological evidence also points to a complex interweaving of different levels of linguistic representation between MSA and dialect (Khamis-Dakwar and Froud 2014). An apparent middle variety thus arises as a result of mixing features from either end of the continuum within a single utterance or stretch of speech. The mixed or middle variety is not a separate attractor in its own right but rather a description of the range of possible points in the middle of the continuum, and the claim that register variation occurs both within and between linguistic levels predicts a potentially infinite number of such points along that continuum. This mixed production was in earlier literature identified as a distinct form ('Educated Spoken Arabic') but is now generally termed 'diglossic mixing' (Owens 2019). The expectation is that linguistic features of different registers will vary on all linguistic levels (i.e., lexicon, syntax, phonology, morphology). The present paper explores whether this is also true of sentence prosody, for the first time. Exploration of this prediction is relevant to the wider study of sentence prosody since Arabic diglossia presents a special case where we may see greater overlapping of prosodic features than seen in bilingual settings.

Another key point for our purposes here is that, although Ferguson argued that the H and L varieties are divergent, in that they have many different linguistic features, he did not claim or expect them to be *discrete*. Indeed, the overlap in features between H and L forms the common ground that underpins the recognition of the two varieties as related. Mejdell (2019) argues for greater attention to the *shared* features between MSA and dialects (cf. also Khamis-Dakwar and Froud 2019) and suggests that these shared features form the background which allows speakers to select distinctive features from either end of the continuum for stylistic purposes. In the present study, we are able to explore, for the first time, which features of sentence prosody, if any, are used this way.

Owens (2019) also notes that most (of the relatively few) prior studies of diglossic mixing in Arabic focus on linguistic features for which the differences between MSA and dialects are well-defined and clear-cut, which presupposes prior descriptions of those features in the literature on Arabic. As we will see in the next section, there are few comparative studies of the prosody of Arabic dialects and descriptions of prosodic differences between MSA and spoken dialects are even more scarce. As a result, it is not surprising that no prior studies of diglossic mixing in Arabic have included prosodic features in the list of variables investigated in their datasets. A search of two of the best quality recent studies, each based on a good volume of data, confirms that in both studies intonation was used solely as a diagnostic for identification of factors affecting other variables of interest; Mejdell (2006) uses intonation to determine whether relative clauses are restrictive or not, in her study of diglossic mixing in Egyptian panel show data, and Hallberg (2016) uses intonation solely to identify clauses as complete or incomplete.

A key aim of the present study is thus to provide the first investigation of register variation in Arabic in which the variables of interest are linguistic features at the sentence prosody level.

### 1.2. Sentence Prosody in Arabic

Work on Arabic sentence prosody has flourished in the last two decades, as evident from the expanding scope of literature summarized in two recent review chapters (Chahal

2006; El Zarka 2017). The majority of spoken Arabic dialects are stress accent languages in which pitch features have a post-lexical function in the form of intonation. The above review articles document a growing number of descriptions of the intonation patterns of individual Arabic dialects in the Autosegmental-Metrical framework (Ladd 2008), which are complemented by earlier descriptions in British School models (Alharbi 1991; Soraya 1966) or using acoustic analysis (Badawi 1965; Rosenhouse 2011). Few studies of intonational variation in spoken Arabic dialects are based on a direct comparison of parallel data. A contributing factor may be the perception of prosodic annotation, on which much analysis of sentence prosody relies, as 'cumbersome' (Watson and Wilson 2017).

Most studies of register variation in Arabic have addressed syntactic, morphological and lexical variation, with some work on phonological variation at the segmental level. It is typically assumed that the prosodic properties of an individual's home dialect will transfer into their formal register (e.g., Benkirane 1998), and this has indeed been documented for some properties such as word-stress placement (Mitchell 1975). A laboratory study of intonational features in formal and spoken Cairene Arabic (El Zarka and Hellmuth 2008) found a greater incidence of secondary accents and shorter prosodic phrases in formal speech, but all other intonational parameters, such as peak alignment, were parallel across the two varieties. There has been no prior work specifically targeting register variation in suprasegmental features including intonation in non-laboratory speech data.

The present study examines register variation between MSA and dialectal San'aani Arabic (SA), spoken in and around the Old City of San'aa, in the capital of Yemen. SA has been described in detail on most linguistic levels except sentence prosody, including syntax, morphology and segmental phonology, by Watson (1993, 1996, 2002). A preliminary description of SA intonation is outlined in Hellmuth (2014). A distinctive feature of SA intonation observed in that study is the use of a rise–fall nuclear contour in information-seeking yes/no questions, in contrast to the rise contour typically observed in the same context in most other Arabic dialects outside North Africa (Hellmuth 2018). The use of a rise–fall contour in yes/no questions is also noted in a preliminary study with Yemeni speakers from the regional city of Taizz (Salem and Pillai 2020).

Another distinguishing feature of SA, shared as an areal feature with other dialects and languages of South Arabia, is utterance-final laryngealization. This term covers a set of related post-lexical phonological processes occurring utterance-finally, before a pause (Watson and Bellem 2011). The key generalizations are that in word- and utterance-final position: obstruents and long vowels are glottalized; oral stops are produced as ejectives; nasals are deleted; sonorants are glottalized, devoiced or deleted; vowels, fricatives and affricates are lengthened (Watson and Asiri 2008). The occurrence of this cluster of properties at the edges of prosodic domains makes laryngealization a potential variable of interest to investigate register variation in MSA–SA.

The choice to focus on register variation in MSA–SA, rather than another MSA–dialect pair, is also facilitated by the serendipitous (if unintentional) elicitation of a sociolinguistic interview recording in which register variation was displayed throughout, which is described in Section 2.1 below.

*1.3. Sentence Prosody and Bilingualism*

Sustained contact between languages in the context of community bilingualism has been shown to result in a range of different effects on the prosody of both first (L1) and second or additional language(s) (L2). The L2 may display prosodic features of a dominant L1 (Nance 2015; O'Rourke 2004), or the L2 may affect the prosody of the L1 (Colantoni and Gurlekian 2004; Fagyal 2005). There are also cases involving the creation of wholly new prosodic features which are properties of neither L1 nor L2 but instead a fusion of the two (Queen 2012), as well as a set of prosodic features which specifically characterize learner intonation (Mennen 2015). These diverse patterns have been argued to be a particular feature of prosody because all languages make use of the same phonetic exponents (pitch, duration and intensity) in some form or other (Bullock 2009). However, there is considerable

variation in the details of the mapping of prosodic form to meaning, both within and between languages, creating an 'indeterminacy' which Sorace (2004) argues is a context that fosters changes to bilingual grammars.

Exploration of sentence prosody and register variation in Arabic is relevant to the wider discussion of sentence prosody in the context of community bilingualism and/or second language (L2) acquisition because of past inference in the literature that MSA is an L2 for Arabic speakers (e.g., Kaye 1972). This assertion was typically based on the fact that MSA is learned in school, thus explicitly, and typically in the context of formal instruction.

Recent evidence suggests characterization of the dialect–MSA relationship as L1–L2 is an oversimplification. Albirini (2019) argues against this claim on the basis of emerging evidence that MSA is acquired implicitly to some extent, by Arabic children growing up in an Arabic-speaking environment, through exposure to media content which is aimed at children and produced in MSA such as cartoons (Albirini 2016). Khamis-Dakwar and Froud (2019) also question the tacit assumption that acquisition of MSA equates solely to literacy development since MSA differs from dialects on many levels of linguistic analysis (alongside many shared features, of course).

However, emerging evidence from neuroimaging studies suggests that dialect–MSA displays patterns of processing which also differ from those seen in balanced bilinguals. In a series of papers, Khamis-Dakwar and Froud (Froud and Khamis-Dakwar 2017, 2021; Khamis-Dakwar and Froud 2014, 2019) argue that L1 Arabic speakers who have grown up in an Arabic-speaking environment show the same type (if not magnitude) of brain response to stimuli in MSA and dialect, which also differs from the brain's response to parallel stimuli in an L2 (such as Hebrew). They call for increased study of dialects and MSA in direct parallel to improve our understanding of the cognitive processing at work in diglossia.

A key point in Ferguson's original proposals is that in diglossia we will observe a markedness relationship between the H and L varieties, in which the H features are a subset of the L features, in particular for phonology. Although this tendency is indeed commonly observed (e.g., an L affix can be added to an H stem, but not vice versa), Owens (2019) reports counterexamples, in the realm of phonology (e.g., dialectal Closed Syllable Shortening applying to an MSA stem); he suggests that future larger scale studies are likely to reveal bidirectional H–L mixing to be the general rule.

This study provides a first opportunity to explore whether there are any indications of a markedness relationship in suprasegmental properties between H and L in domains larger than the word. The primary hypothesis of the study, however, is that a complex interweaving of features, which is the hallmark of diglossic mixing on other levels of linguistic analysis, will be found also in sentence prosody.

### 1.4. The Present Study

The present study examines three variables operating at the level of sentence prosody: (i) F0 variation, within and between turns at talk; (ii) intonational choices, including the type and distribution of pitch accents and phrase boundaries; (iii) incidence of utterance-final laryngealization. These variables are investigated in data from a single speaker, whose utterances are first coded for register on three levels: formal or *fusħa* (F), middle or *wusṭaː* (W) and dialect or *ʕaːmijja* (A). The coding is based on non-prosodic features to avoid circularity, generating 400+ turns for analysis. Owens (2019, p. 89) laments the lack of large-scale studies of diglossic mixing in Arabic but acknowledges the difficulty in eliciting or obtaining the data needed for larger studies. He further notes that the many existing small case studies, despite their limitations in size, are nonetheless valuable for generating hypotheses to explore in larger studies, and for identifying variables of interest to investigate further. To the best of my knowledge, this is the first study of diglossic mixing of sentence prosody in any MSA–dialect pair, but it is certainly the first to investigate sentence prosody in the context of MSA–SA mixing. This study also serves as a

potential model of methods for the investigation of sentence-level prosody across registers of Arabic.

## 2. Methods

### 2.1. Participants, Materials and Procedure

The data examined are from a single speaker (f2) in a 20 min sociolinguistic interview. Two female participants (f1/f2) took part, with the author as the interviewer. The participants are sisters, aged 20–25 years at the time of recording, recruited through personal contacts of the author. The participants are from the Al-Ga'a district, adjacent to the Old City of San'aa; their extended family originate from a village in Greater San'aa (name redacted for anonymity). The author/interviewer has British English as L1 and learned Arabic as an adult largely in formal educational settings. The interview is part of a small corpus of data collected in San'aa in 2008. Participants provided informed consent to record audio of their speech and to use the transcripts and data excerpts in research (but not to open access sharing of the audio recording, due to cultural sensitivities).

The sociolinguistic interview was conducted using the Sense Relation Network (SRN) tool (Llamas 2007). The SRN is designed to encourage participants to use their vernacular speech variety by inviting them to discuss dialect-specific lexical choices in a conversation with another member of the same speech community. A version of the SRN was created to target lexical items reported to vary between the variety spoken in the Old City of San'aa and other Yemeni dialects (Watson 1993, 1996, 2000).

The interview took place in a quiet office. The audio was recorded directly to wav format at 44.1 KHz 16 bit using a Marantz PMD660 recorder. Each participant was recorded to a separate channel in the stereo file, via a Shure SM10 headset microphone.

The SRN is an elicitation tool rather than an experimental 'word list' task. Target words and phrases are presented on a network diagram as prompts for spontaneous verbal discussion between a pair of participants about the lexical items they use in their variety. In Arabic, this involves the presentation of target words in written MSA, for the list of target meanings (i.e., 'senses') for which participants are invited to report their local variants. Of the two participants in this interview, only f2 was sufficiently confident in reading MSA to work directly from the text prompts. Speaker f2 thus took the lead in directing the conversation, liaising between her two interlocutors whose familiarity with different varieties of Arabic varied greatly: speaker f1 was an expert L1 speaker of SA whereas the author/interviewer was an L2 speaker of Arabic with relatively limited exposure to SA, but good fluency in other dialects/varieties of Arabic. Speaker f2's talented and sensitive navigation of this linguistic situation led to considerable intra-speaker register variation throughout the conversation, which forms the basis of this case study.

### 2.2. Analysis

Transcription: The interview data was manually segmented into turn-sized sections, typically mapping to one or two Intonational Phrases (IP). Each turn was orthographically transcribed by the author in ELAN (Sloetjes and Wittenburg 2008) using a phonetically transparent roman alphabet transliteration system for Arabic, devised by Hellmuth and Almbark (2019). The stereo wav file was split to extract the audio signal for each speaker, in Praat (Boersma and Weenink 1992–2018), and the text transcription was force-aligned to the mono audio file using Prosody Lab Aligner (Gorman et al. 2011) as an aid to later coding and annotation. The resulting mono sound file and aligned Praat TextGrid for speaker f2 (only) were then used for further analysis.

Coding: Each turn produced by f2 was coded for the register of Arabic on three levels: *fusħa* (F), *wusṭaː* (W) and *ʕaːmijja* (A). These levels correspond to Mitchell's (1984, 1986) formal/careful/casual distinction but were defined and coded in the present study according to a specific set of criteria. The decision to code with only three levels was made for pragmatic reasons; the data displays consistent mixing of linguistic features from different registers within and between turns, and it would not have been possible to

determine, a priori, which constellations of features correspond to which level(s). Since the aim of the present study is to determine which features of sentence prosody participate in diglossic mixing, the coding in this study was performed with reference to lexical choices, morphology and segmental phonology only; some examples are shown in (1–2).

1.  a.  F   naˈquːlu           ˈla-haː
            say.1PL            to-it
    b.  W   naˈɡuːlu           ˈla-hu
            say.1PL            to-it
    c.  A   nuˈɡulluh
            say.1PL.to.it
            'we say for it'

2.  a.  F   laˈdaj-naː
            to-us
    b.  W   laˈdeː-na
            to-us
    c.  A   ˈʕinda-na
            to-us
            'we have'

In (1) we see the same lexical item in all three registers (the root <قال> [q-a-l] 'to say') but differences between F/W versus A in morphology, with the prepositional clitic affixed directly to the verb in A only; in contrast, we see a difference between F versus W/A in the segmental phonology in the realization of the target sound [q] <ق> 'qaf', with [q] in the F register but [ɡ] in both W and A. In (2) we see different lexical choices in F/W versus A, with the distinction between F and W indicated through monophthongization of [aj] to [eː] in W only. Further codes were used to indicate turns produced in English (E) or where the content was uninterpreted, e.g., a hesitation marker (U).

All turns in the data were coded by the author at two time points more than one year apart, and by a second coder who is a first language speaker of Arabic. Inter-code agreement (between either of the two author codes and those of the independent coder) was initially 60% (327/548). The remaining data were discussed and the majority of differences arose from the treatment of mixed turns (where part of the turn was in one register and part in another). A 'whole clause' approach was thus applied: the register of the majority of information in a turn was applied to the whole turn, even if it contained an isolated word or phrase with features of a different register (an example will be seen in Figure 6 below). Any remaining discrepancies were resolved by discussion to reach a consensus. The final coding involved some adjustments to turn boundaries, yielding a final turn count of 469.

Annotation: The wav file and TextGrid were segmented into turn-sized short files, and then each turn was prosodically annotated by the author and labelled for the presence/absence of the post-lexical phonological process of turn-final laryngealization. Prosodic annotation was performed following the conventions of the Autosegmental-Metrical framework (Ladd 2008), using the putative 'language-neutral' tone label set proposed by Hualde and Prieto (2016). The use of this language-neutral annotation label tagset results in minor differences in the annotation here of tunes previously discussed in Hellmuth (2014) but none of these minor differences are at issue in the examples discussed below. The adopted inventory of tone labels assumes two levels of phrasing (intermediate and intonational phrases) and thus includes pitch accents (marked '*'), phrase accents ('-') and boundary tones ('%'). A stylized representation of the pitch contour for each pitch accent label is provided in Appendix A.

Categorical presence or absence of turn-final laryngealization was identified from auditory impression with reference to the spectrogram and waveform in Praat, with comparison to detailed descriptions of SA turn-final laryngealization in different phonological contexts (Watson and Asiri 2008). To control for phonological context in the analysis, the syllable type for each turn-final word was recorded during annotation e.g., CVVT [ṭariːɡ]

'path'; CVVN [tama:m] 'fine, okay' (where T stands for 'any obstruent' and N stands for 'any nasal'). A Praat script was used to extract annotation labels from each turn level TextGrid, along with a count of the number of words in each turn.

F0 measurement: A Praat Pitch object was created for each turn (using default settings). All turns coded as F/W/A were inspected and manually corrected for tracking errors. A Praat script was used to extract the following F0 measures from each corrected Pitch object, in Hz and semitones: minimum, maximum, mean, standard deviation (SD) and median; the maximum and minimum were then used to calculate the F0 range in octaves [log2 (maxF0/minF0)] for each turn.

Data visualization and statistical analysis: The descriptive results of each layer of analysis were visualized using ggplot2 (Wickham 2010) supported by further exploration of acoustic data using linear regression models run in R (R Core Team 2014); mixed models with random effects were not appropriate as the data do not involve repeated measures.

## 3. Results

### 3.1. Overview of the Data

The data comprise 469 turns: 35 were coded as uninterpretable (e.g., hesitation markers) and 5 were produced partially or in whole in English, and these were excluded leaving 429 for analysis. The split of codes for the remaining data was: F:N = 44 (10%); W:N = 200 (47%); A:N = 185 (43%). Figure 1 shows the number of turns by register (1a) alongside a count of the number of turns of each length (by word count) in each register (1b). Figure 2 visualizes the distribution of each register type along the timeline of the 20 min interview, generated using vistime (Raabe 2021).

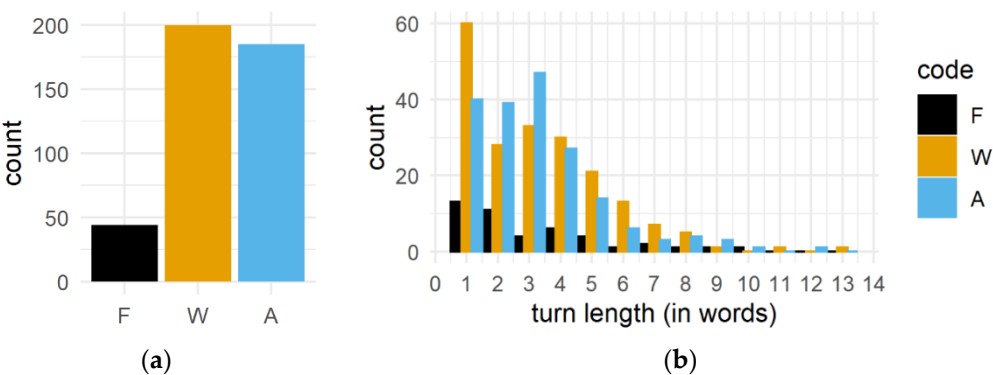

**(a)** **(b)**

**Figure 1.** (**a**) Count of turns coded in each register type; (**b**) distribution of turn lengths in words, by register: *fusħa* 'formal' (F), *wusṭa:* 'careful' (W) and *ʕa:mijja* 'casual' (A).

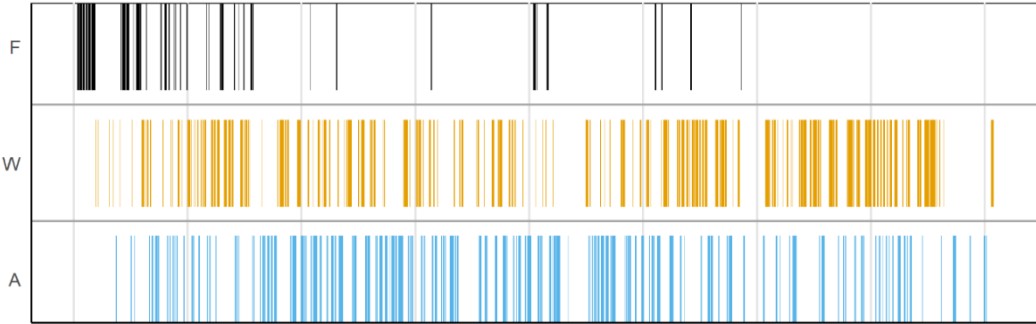

**Figure 2.** Distribution of turns in interview timeline by register: *fusħa* (F)/*wusṭa:* (W)/*ʕa:mijja* (A).

Figures 1 and 2 show that the formal register was used in only a small proportion of the data and mostly at the beginning of the interview (in the first 4–5 min). Although the presentation of the target lexical items in written MSA initially elicited speech in formal register, the interactive nature of the SRN tool was successful in encouraging speaker f2

to gradually move towards the use of dialectal forms. In Figure 2 we can see that the careful register (W) is initially used to replace the formal (F) register, with the use of fully dialectal speech (A) following shortly afterwards; from about 5 minutes onwards, speaker f2 is largely using either W or A. Continued use of both registers was probably due to the presence of a non-vernacular speaker as the interviewer (favouring the use of the careful register, W), balanced against a shared focus on local lexis (favouring the use of the casual register, A).

Although fewer turns were produced in F than in W/A, the mean turn length in words is similar in all three registers (F = 3.14; W = 3.21; A = 3.11). The high number of single word turns coded as W is due to the decision to code all instances of the single word turn [tama:m] 'okay' (N = 41) as W (see discussion in Section 3.4). A data subset without these turns is used in relevant parts of the analysis (N = 388; F = 43 (11%); W = 160 (41%); A = 185 (48%)).

### 3.2. F0 Variation

Table 1 reports the mean and SD for the F0 measure by register code. The spread of values for these F0 measures across turns, by code, is illustrated in Figure 3.

**Table 1.** Mean (standard deviation) of measures of F0 variation across turns, by register code.

| Register | Min (Hz) | Max (Hz) | Mean (Hz) | SD (Hz) | Median (Hz) | Range (Octaves) |
|---|---|---|---|---|---|---|
| F | 215.58 (15.35) | 329.68 (32.51) | 258.97 (17.17) | 29.34 (9.27) | 251.84 (15.20) | 0.61 (0.14) |
| W | 212.41 (21.53) | 318.89 (35.86) | 256.64 (18.84) | 27.09 (9.80) | 252.00 (19.79) | 0.58 (0.20) |
| A | 209.11 (19.83) | 325.53 (48.69) | 261.74 (29.29) | 29.77 (14.43) | 259.46 (31.85) | 0.63 (0.23) |

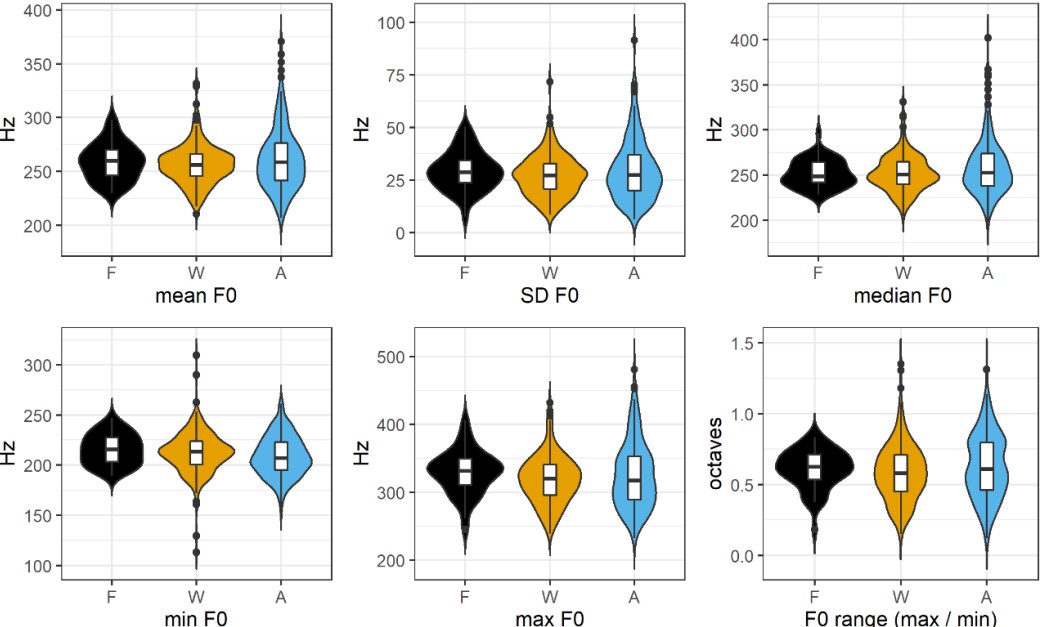

**Figure 3.** Median and interquartile range and frequency distribution of values across turns by register code for mean, SD, median, min and max values of F0, and F0 range (max/min) in octaves.

These measures reveal subtle differences only in the degree of F0 variation across registers. A wider range of variation is visible in turns labelled W or A, than F, but this is largely attributable to the larger number of tokens for those codes (90% of the data).

A series of linear regression models were run to predict each F0 measure in turn as the dependent variable, as a function of register *code* (e.g., minf0Hz~code) with treatment coding (i.e., with one level of the factor *code* as reference level); the model was re-run after re-levelling *code* to a different reference level to obtain pairwise comparisons. The only significant differences found in measures of F0 variation across registers were between W and A: median F0 is lower in W than A ($\beta = -7.458$; SE = 2.589; t = $-2.88$; $p = 0.0042$); mean F0 is lower in W than A ($\beta = -5.106$; SE = 2.427; t = $-2.104$; $p = 0.036$); SD of F0 is lower in W than A ($\beta = -2.674$; SE = 1.22; t = $-2.188$; $p = 0.029$); F0 range in octaves is narrower in W than A ($\beta = -0.0432$; SE = 0.021; t = $-2.034$; $p = 0.043$). There were no significant differences in measures of F0 variation between W and F.

The overall similar range of F0 variation across registers is perhaps to be expected as these are data from a single speaker and thus reflect her individual pitch range. The observed differences indicate greater use of higher and/or more expanded pitch by speaker f2 in A than W. The distribution of F0 range values is slightly bimodal in the A register, indicating a split which is also visible to a lesser extent in the distribution of values of max F0 in the A register. This split reflects the fact that f2 produced a subset of A-coded turns in a much wider pitch range, which have the auditory impression of being 'performed', as an example for the interlocutor of how an utterance would be produced naturally in context between SA speakers. Figure 4 shows an example in which f2 provides a sample of how a SA lexical item (['ɡawħaza] 'to sit') would be used; the reporting clause ('And she says:', coded W) is produced in a relatively narrow pitch span (0.5 octaves), but the reported clause (coded A) is produced in a very wide pitch span (1.3 octaves).

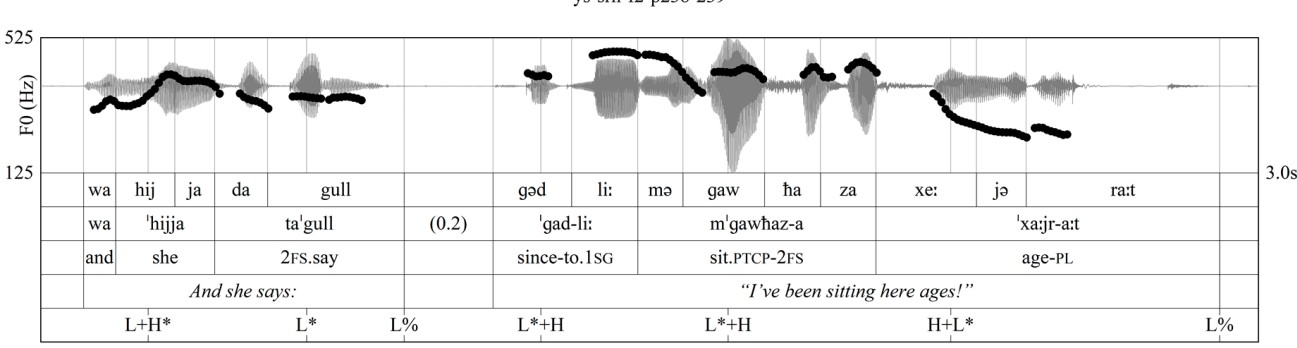

**Figure 4.** Sequence of turns (W then A) with narrow versus wide pitch span (0.5 versus 1.3 octaves).

In summary, then, the data reveal greater pitch variation in the casual (A) register than in the careful (W) and formal (F) registers. F0 variation is thus a linguistic feature relevant for the investigation of diglossic mixing in Arabic, as shown also for measures of F0 variation in formal versus informal speech in other languages such as Korean (Winter and Grawunder 2012). In the present data, however, the pattern observed in reported clauses suggests the variation here may be a by-product of differences in the semantic and/or pragmatic content expressed rather than an inherent property of any one register.

### 3.3. Intonational Phonology

Table 2 shows token counts for all pitch accent labels by register and Figure 5 illustrates the distribution of pitch accent types by register. The inventory of pitch accents used to label the F register data forms a subset of those needed to label the W/A registers. All registers share the property of using L* and H* as the most frequent pitch accents, with some use of bitonal rising pitch accents in all three also (L+H* and L*+H), but, bitonal falling pitch accents are used in W/A only, and the H+!H* pitch accent is more frequent in A than W.

**Table 2.** Token count of all pitch accent labels, by register code.

| Register | L* | H* | !H* | L+H* | L*+H | H*+L | H+L* | H+!H* |
|---|---|---|---|---|---|---|---|---|
| F | 66 | 28 | 5 | 14 | 3 | 0 | 0 | 0 |
| W | 246 | 144 | 56 | 33 | 15 | 11 | 3 | 7 |
| A | 170 | 112 | 58 | 32 | 37 | 28 | 3 | 36 |

**Figure 5.** Observed pitch accent types as a percentage of total pitch accent tokens, by register.

Table 3 shows token counts for all edge tone labels, by register code. Most of the variation in the count of edge tones is due to the different volumes of data in each register. A count of the number of non-turn-final edge tones (phrase accents/boundary tones combined), as a proportion of the number of multi-word turns per register, in fact, reveals little difference in phrasing patterns between registers, as shown in Table 4. This is of note since differences in the distribution of phrase boundaries were reported as a feature of register variation for speakers from Egypt (El Zarka and Hellmuth 2008).

**Table 3.** Token count of all edge tone labels, by register code.

| Register | L- | H- | !H- | Total | L% | H% | !H% | Total |
|---|---|---|---|---|---|---|---|---|
| F | 1 | 19 | 0 | 20 | 13 | 31 | 6 | 50 |
| W | 25 | 48 | 0 | 73 | 108 | 114 | 0 | 222 |
| A | 25 | 47 | 3 | 75 | 144 | 51 | 12 | 207 |

**Table 4.** Incidence of turn-internal phrasing boundaries, by register.

| Register | Turns with > 1 Phrase | Turns with > 1 Word | % |
|---|---|---|---|
| F | 15 | 31 | 48% |
| W | 74 | 140 | 52% |
| A | 74 | 145 | 51% |

Tables 5 and 6 show the distribution of 'simple' and 'complex' nuclear contours, respectively, by register, including all observed pitch accent boundary tone combinations. Cells of the table which account for 10% or more of the turns for that register are shaded in grey, in both tables. The 'simple' contours make up 98% and 91% of turns in the F/W registers respectively, but only 76% of turns in the A register.

**Table 5.** Observed 'simple' nuclear contours, as a percentage of all turns in that register.

| Register | L* L% | L* H% | L* !H% | H* H% | H* L% | !H* L% |
|---|---|---|---|---|---|---|
| F | 16% | 56% | 14% | 0% | 5% | 7% |
| W | 17% | 43% | 0% | 5% | 10% | 18% |
| A | 21% | 15% | 4% | 3% | 11% | 21% |

**Table 6.** Observed 'complex' nuclear contours, as a percentage of all in that register.

| Register | L+H* L% | L+H* H% | L+H* !H* | L*+H H% | H*+L L% | H*+L H% | H+!H* L% | H+!H* H% | H+L* L% |
|---|---|---|---|---|---|---|---|---|---|
| F | 0% | 2% | 0% | 0% | 0% | 0% | 0% | 0% | 0% |
| W | 1% | 1% | 1% | 0% | 1% | 2% | 4% | 0% | 1% |
| A | 0% | 1% | 1% | 1% | 5% | 0% | 13% | 3% | 2% |

This apparent difference in the complexity of contours between the F/W versus A ends of the register continuum is largely driven by the high incidence of the H+!H* pitch accent in A-coded turns. Figure 6 shows an example of the distinctive H+!H* contour seen in many A-coded turns. Although the register coding was performed based on lexical and segmental features, the second transcriber remarked that many turns (which were later annotated with H+!H*) stood out as having 'Yemeni intonation'.

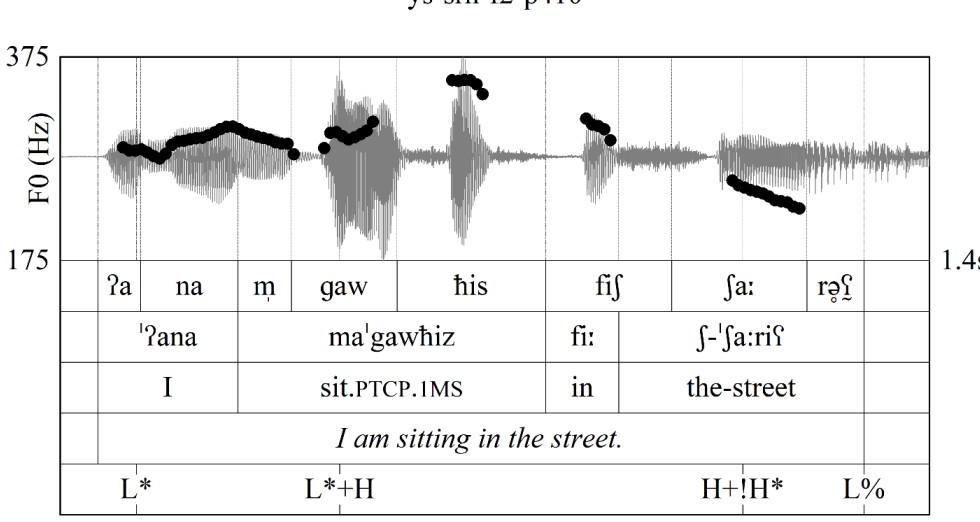

**Figure 6.** An A-coded turn realized with an H+!H* L% nuclear contour.

Another tendency in the data is a lower proportion of falling contours in the F register, in comparison to W/A, which may be due to speaker f2's realization of many F turns as a sequence of short phrases, each of which bears a continuation rise, followed by a very short final phrase, in a pattern commonly heard in broadcast MSA speech; other patterns reported in broadcast MSA, such as sequences of early peak falls (Rastegar-El Zarka 1997), are not seen in the present data. Figure 7 shows an example of an F-coded turn realized with a series of continuation rises on short phrases, followed by a very short final phrase realized in a compressed pitch range.

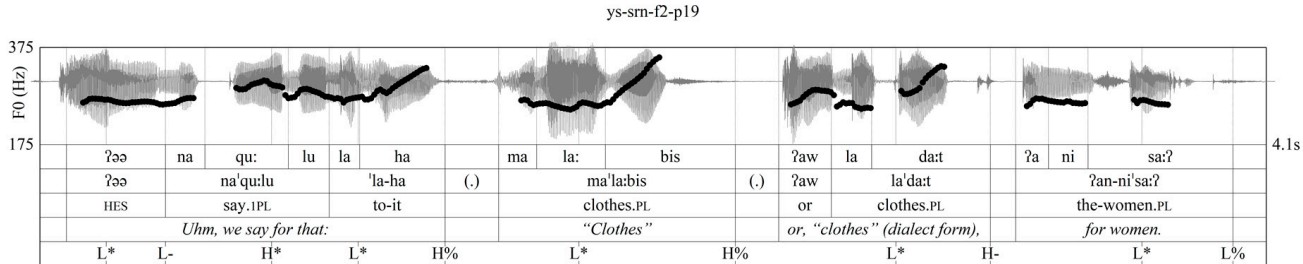

**Figure 7.** An F-coded turn with continuation rises and a broadcast MSA-style final short phrase.

One further feature that was shared across all three registers was the occasional use of secondary accents, whereby a word is realized with two pitch accents: one on the stressed syllable as expected, but another also on another syllable earlier in the word. The use of secondary accents in MSA but not dialectal speech was observed in a laboratory study of each register as produced by the same Egyptian speakers (El Zarka and Hellmuth 2008). In the present study, secondary accents are rare, but are more common in the F/W registers (three examples each): F: [ˌʕaːˈmijja] 'dialect' (turn 4); [ˌar-riˈʒaːl] 'the-man' (turn 9); [ˌtalafazˈjoːn] 'television' (turn 324); and W: [ˌal-ʔaˈðaːn] 'the-ears' (turn 84); [ˌtalafazˈjoːn] 'television' (turn 329); [ˌal-ʕaˈṣiːd] 'dumpling' (turn 447)). There is just one example in an A-coded turn (A: [ˌgusguˈsiː] 'puppy' (turn 354). An example from a W-coded turn is shown in Figure 8.

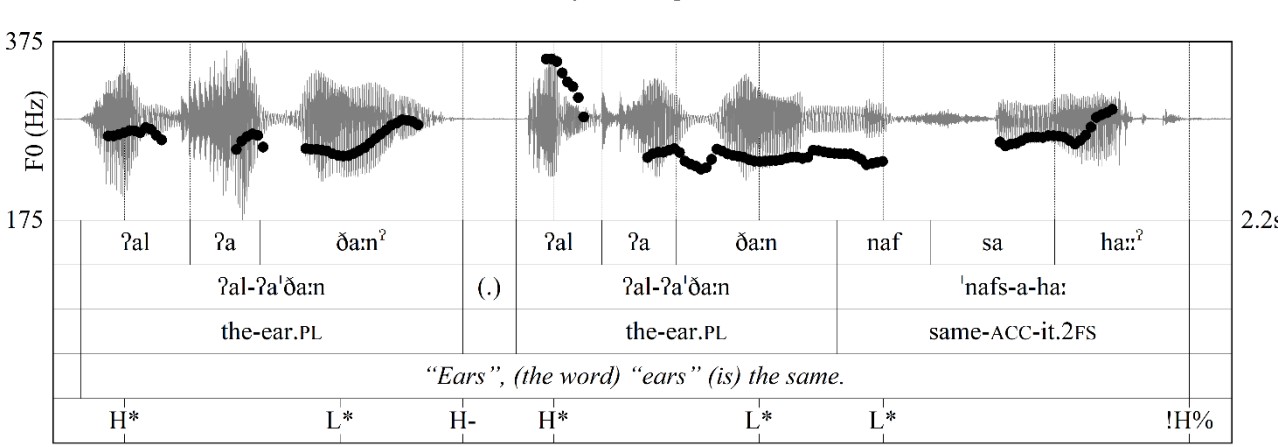

**Figure 8.** A W-coded turn showing secondary accents on the word [ˌal-ʔaˈðaːn] 'the-ears'.

In summary, the three registers share a core common inventory of pitch accents but falling bitonal pitch accents were only used in W-/A-coded turns, and more frequently so in the A register (particularly H+!H*). This difference contrasts with an F/W versus A distinction in the relative 'complexity' of nuclear contours. The incidence of turn-internal phrasing boundaries was similar across registers, but although all three registers contained examples of secondary accents they were more common in F/W than in A.

*3.4. Post-Lexical Laryngealization*

The proportion of turns in which laryngealization was identified in the final lexical item varied by register code: F had laryngealization in 18 out of 44 turns (41%); W in 98 out of 200 (49%), but A in 149 out of 185 (81%). Speaker f2 thus produces utterances with final laryngealization to an increasing extent as she moves from formal to dialectal speech. We might argue from this overall result that F/W pattern together in showing relatively low levels of laryngealization, in contrast to A where the rate is much higher. However, it is necessary to control for internal (linguistic) factors which also influence the incidence of laryngealization; the relevant factors in SA are the manner of articulation of the final consonant(s) and syllable structure (Watson and Asiri 2008).

Figure 9 shows the proportion of laryngealization for the most commonly observed syllable shapes (N = 400), by register code. The pattern in A-coded turns is of near categorical laryngealization of utterance-final obstruents and non-nasal sonorants, but slightly less of nasals; words ending in open syllables—which never attract stress—undergo laryngealization much less, consistent with Watson and Asiri's (2008) observation that unstressed final syllables are less likely to be reduced. W-coded turns display a similar pattern of sensitivity to stress and final consonant (reduced incidence in CV open syllables and final nasals). The number of data points for F-coded turns is small, but we can see a

contrast in the treatment of CVV versus CVVN syllables, with the former much more likely to be laryngealized than the latter, matching the pattern in the A/W- coded turns.

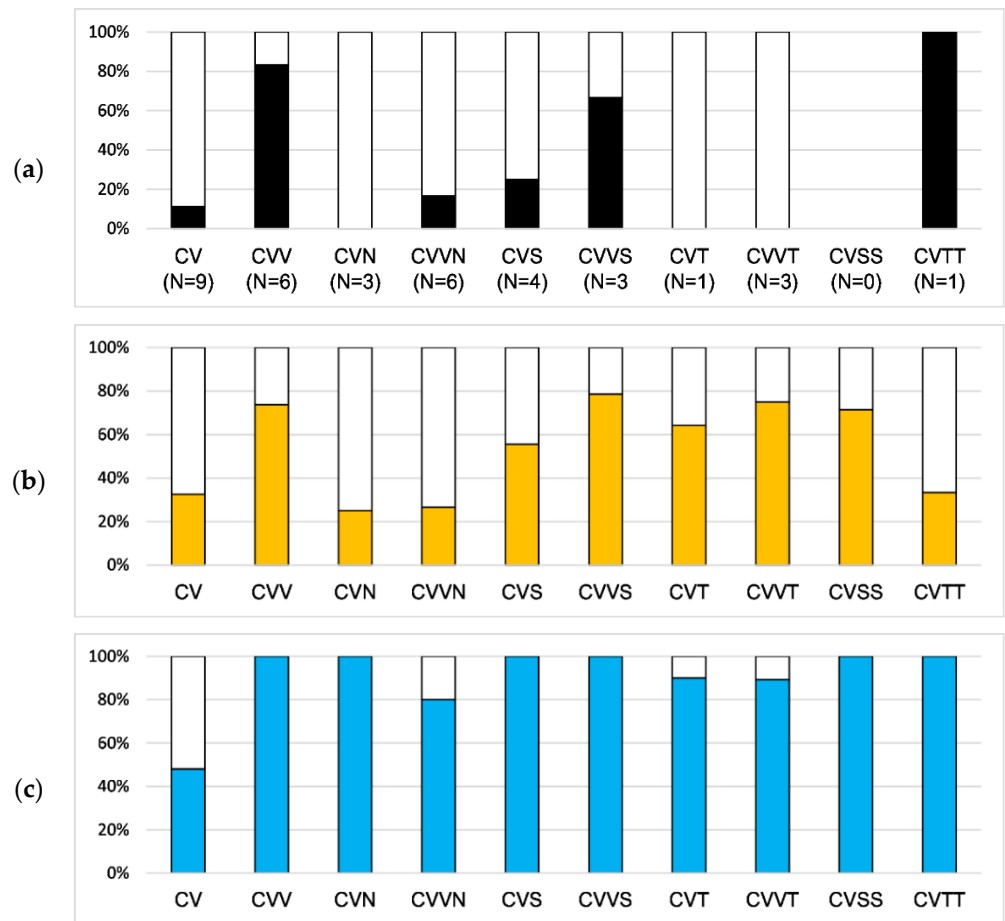

**Figure 9.** Proportion of turns displaying laryngealization by syllable type/shape and by register for (**a**): *fusħa* 'formal' (F); (**b**): *wusṭaː* 'careful' (W); and (**c**): *ʕaːmijja* 'casual' (A). Key to syllable codes: C: any consonant; V: any vowel; N: any nasal; S: any non-nasal sonorant; T: any obstruent.

Overall then, the A-coded data show the expected patterns of laryngealization for a speaker of the SA dialect. The same speaker displays much less use of laryngealization in turns coded as F/W, but with some evidence of similar phonological conditioning to that observed in A-coded tokens.

There is an indication in the data that these patterns are under the speaker's control. Figure 10 shows self-repair by speaker f2 of application of laryngealization on the word /jafˈʕal/ 'do.IMPF.3MS' (realized the first time as [jafˈʕaːlˀː]) at the potential completion point of a turn realized with F features (and where f2 is producing verbatim a text prompt written in MSA). She immediately produces an increment to the turn which is realized with the same prosodic contour as the host phrase which it repairs, except for suppression of laryngealization on the utterance-final word. Figure 11 illustrates the phonetic detail of the realization of the minimal pair realizations of the phrase-final word.

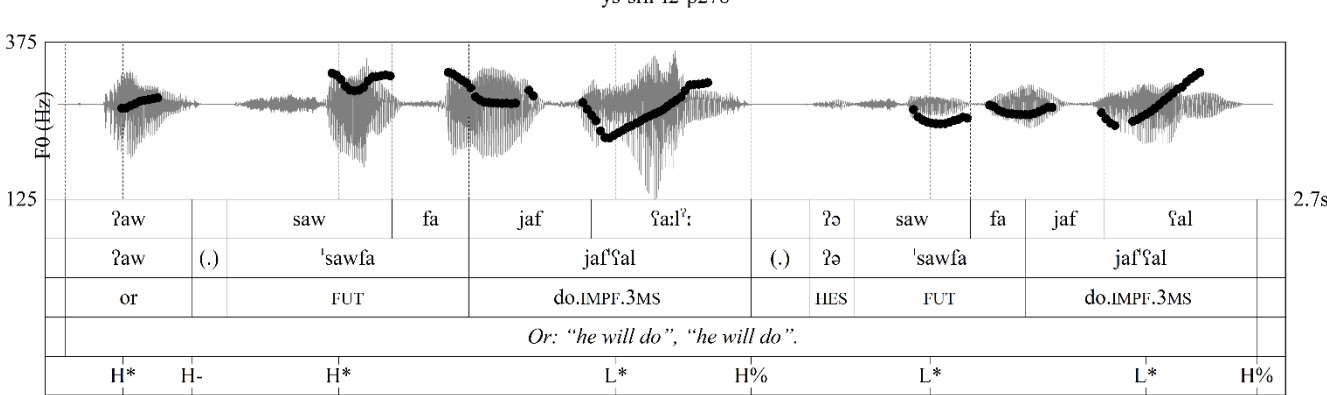

**Figure 10.** Sequence of F-coded turns produced with and without utterance-final laryngealization.

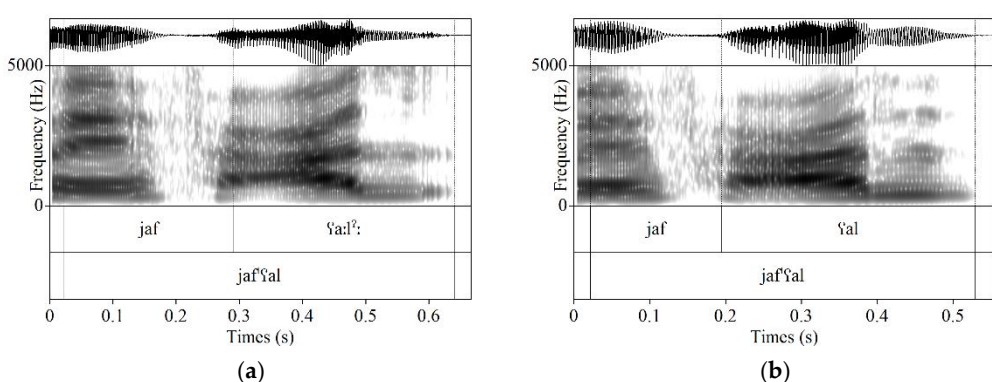

**Figure 11.** Two instances of [jaf.ˈʕal] (from Figure 10) with (**a**) and without (**b**) laryngealization.

Finally, there is some evidence also of co-variation between laryngealization and choice of prosodic contours. A large number of single word tokens of the discourse marker [tamaːm] 'okay/fine' were produced by speaker f2 throughout the interview (N = 41). This word is a viable lexical item in both W or A registers, and does not display phonological features specific to W or A either; all tokens were coded as W, as in the interactional context of the sociolinguistic interviewer the intended audience was more likely to be the interviewer (for whom W is accessible) rather than the other participant. These 41 tokens vary in the incidence of laryngealization and also in the choice of nuclear contour, as set out in Table 7. The majority of [tamaːm] turns are realized without laryngealization (78%), and the same proportion of turns are realized with a 'simple' rise contour (L* H%, also 78%). Although these choices do not strictly co-vary, the overall pattern is of a tendency to produce the discourse marker with prosodic features which fall in the common ground between F and A (since both registers use L* H%) but towards the formal end of the continuum of variation (and thus without SA dialectal laryngealization).

**Table 7.** Co-variation in laryngealization and choice of nuclear contour in tokens of [tamaːm] 'okay'.

| Laryngealized? | L* H% | L* !H% | H* L% | L+H* L% | H*+L L% | Total |
|:---:|:---:|:---:|:---:|:---:|:---:|:---:|
| yes | 5 | 0 | 2 | 2 | 0 | 9 |
| no | 27 | 1 | 3 | 0 | 1 | 32 |
| Total | 32 | 1 | 5 | 2 | 1 | 41 |

Interestingly, the only contour which does co-vary with the presence of laryngealization is L+H* L%, which is the contour typically observed in information-seeking yes/no-questions in SA (Hellmuth 2014). In these utterances we might conjecture that the intended audience of the turn was f2's fellow participant (for whom A is accessible) rather than

the interviewer, leading to its realization in the A register and with SA prosodic features. Figure 12 shows an example of one of these [tama:m] turns, alongside a minimal pair realization of the same word with an MSA yes/no-question contour (L* H%).

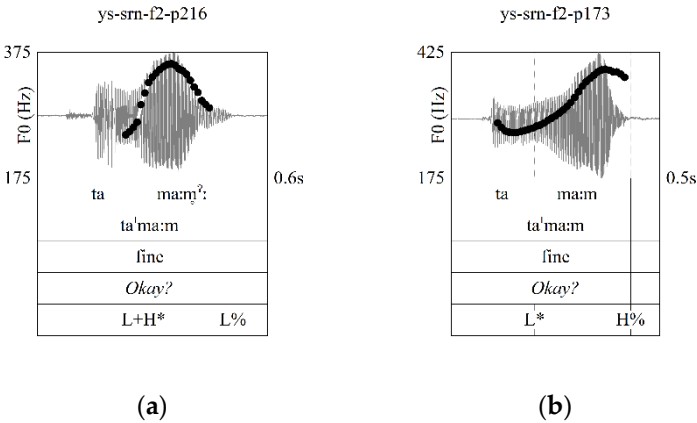

(**a**)                                    (**b**)

**Figure 12.** Single word W-coded turns of [tama:m] realized (**a**): with SA yes/no-question contour and with laryngealization; (**b**): with MSA yes/no-question contour and no laryngealization.

## 4. Discussion

Table 8 summarizes the observed differentiation of the identified registers of Arabic produced by speaker f2 in this case study data. The results show an interweaving of the use of different aspects of sentence prosody across the three registers of speech, with at least one feature serving as a cue to each of the possible ways of grouping the registers, though no feature fully differentiates F vs. W vs. A. All three registers shared the same density of phrasing boundaries, but the pattern of using a sequence of continuation rises on short phrases within a turn was a hallmark of F-coded turns only. A number of features distinguish A from W/F, and these features span all three of the investigated variables.

**Table 8.** Summary of observed variation in differentiation of registers, by prosodic feature type.

| Pattern | F0 Variation | Intonation | Laryngealization |
|---|---|---|---|
| F&W vs. A | bimodal median F0 level of median F0 | pitch accent inventory size use of secondary accents broadcast MSA-style contours | % laryngealization |
| F vs. W&A | – | | – |
| F vs. W vs. A | – | – | – |
| F&W&A | – | density of phrasing boundaries | – |

The observed larger pitch accent inventory in A is a potential example of a markedness relationship between H and L predicted by Ferguson (1959). However, none of the features which distinguish A (in the top row of the table) are categorically exclusive to A. For example, W-coded turns also displayed a tendency towards bimodal median F0 and contained some tokens of falling bitonal pitch accents; also, some laryngealization was seen in all registers, and there was one example of a secondary accent in an A-coded turn.

This mixing of features across registers is consistent with the characterization of Arabic diglossic mixing as an interweaving of features of both A and F along a continuum of variation. The present dataset is small and limited in interactional scope, but there was some evidence here of the speaker displaying control over this variation, in the example of self-repair when A features were used in an otherwise F-framed turn. This self-repair was of a word produced with L features in H context—thus a counterexample of the type Owens (2019) cites as evidence of bidirectional mixing—but is repaired by speaker f2.

The present results for MSA–SA reveal a general picture of shared prosodic features across registers, alongside distinct features which serve to differentiate the registers at each end of the continuum, at least some of which appear to be under the speaker's control. More work is needed to expand the volume of data, discourse context types and number of speakers investigated, but this study has identified variables and methods of analysis that can be used in future studies to further explore the role of sentence prosody in register variation in Arabic.

**Funding:** This research was funded by the University of York Research Priming Fund (Project Name: Prosodic Variation in Arabic). The APC was funded by the Department of Language and Linguistic Science, University of York.

**Institutional Review Board Statement:** The study was conducted in accordance with the Declaration of Helsinki, and the protocol was approved by the University of York Humanities and Social Sciences Ethics Committee (January 2008, Project Name: Prosodic Variation in Arabic).

**Informed Consent Statement:** Informed consent was obtained from all participants in the study.

**Data Availability Statement:** Derived data (such as measurements) are available from the author on request; the study participants did not consent to third party sharing of their audio data.

**Conflicts of Interest:** The author declares no conflict of interest.

**Appendix A**

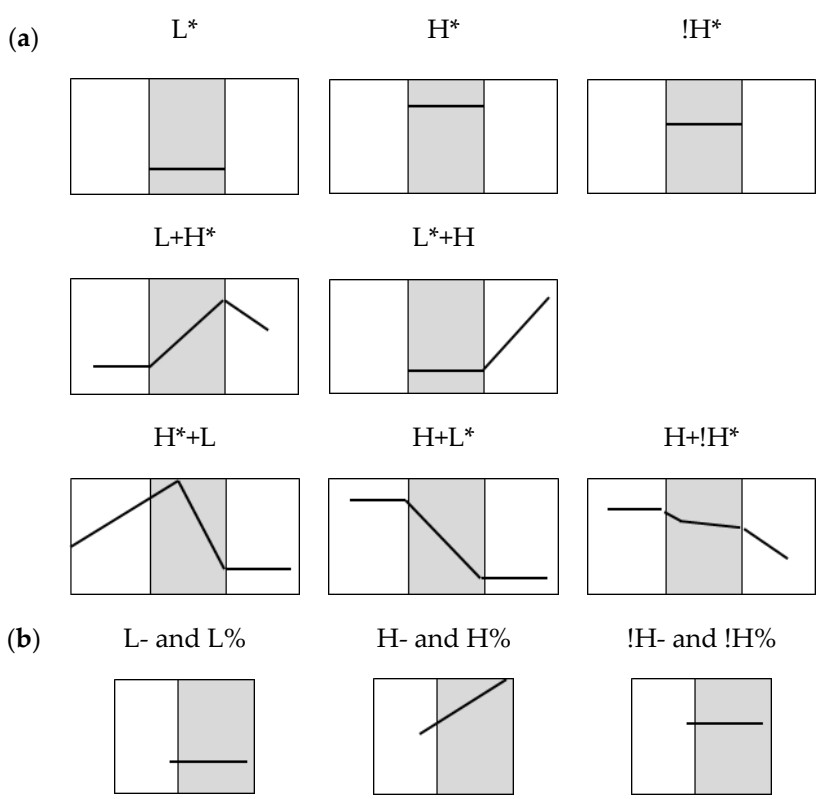

**Figure A1.** Schematized representation of a typical pitch contour labelled for (**a**) pitch accents and (**b**) edge tones. Boxes represent syllables; for pitch accents, the shaded box indicates the position of the accented syllable; for edge tones, the shaded box indicates the last syllable in the intermediate or intonational phrase.

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
