# Peer review of "Sentence Prosody and Register Variation in Arabic"

_languages, doi:10.3390/languages7020129_

Round 1

Reviewer 1 Report

please cf. attached pdf file for the review

Author Response

Please see the attachment,

Reviewer 2 Report

This is an excellent, original, important paper. To the best of my knowledge, it is the first study of the role of sentence prosody in register variation in Arabic. The variables are carefully chosen on the basis of previous literature. This study could be replicated with a larger number of speakers - the only hesitation I have about the study is that it is based on a 20-minute sociolinguistic interview of a single speaker; however, the methods are clearly replicable, data visualisation is good and appropriate, and the findings predictable from earlier work. I found one typographical error: 'We might argue from this overall result that F/A pattern together in showing relatively 466
low levels of laryngealization, in contrast to A where the rate is much higher.' F/A should be F/W, I think.

Author Response

Please see the attachment,

Reviewer 3 Report

Languages review

This paper examines register (High vs Low) variation in one speaker of San’ani Arabic, measuring three acoustic characteristics.

Crucially, the coding of turns as H or L was not based on the same variables that were being measured in the study. The results showed different median f0 across registers, while also showing mixing of features across registers. This work is an addition to the literature since prosodic features of this diglossic mixing have not been examined before, as well as adding to the literature on prosody in Arabic varieties.

Overall, this study is well conducted and the analysis appears reliable and valid.

Specific comments are below.

Introduction

Since this study examines f0 in different registers, it may be relevant to add some literature on this question in other languages, such as the following articles:

Loveday, L. 1981. Pitch, politeness and sexual role: An exploratory investigation into the pitch correlates of English and Japanese politeness formulae. Language and Speech, 24, 71-89.

Winter, B. & S. Grawunder. 2012. The phonetic profile of Korean formal and informal speech registers. Journal of Phonetics, 40(6), 808-815.

On p5 the description of the interview notes that there were two other participants – could the author expand on any possible influences this may have had on the subject’s productions? – including perhaps more detail about the interviewer’s language background.

Results

The figures are clear and informative.

p.8 - In Table 1 it looks like the column for Max (Hz) is incorrect – for each register, the number in this column is the same as for Min, which doesn’t make sense and doesn’t line up with the figures. Probably just a copy-paste issue.

p.9 – just a clarification qs about the linear regression: since Register has 3 levels (F,W,A), wouldn’t it be the case that some kind of post-hoc pairwise test would need to be conducted to determine which registers were different from one another? This is not mentioned.

Author Response

Please see the attachment,
